# TERT Promoter and BRAF V600E Mutations in Papillary Thyroid Cancer: A Single-Institution Experience in Korea

**DOI:** 10.3390/cancers14194928

**Published:** 2022-10-08

**Authors:** Min Jhi Kim, Jin Kyong Kim, Gi Jeong Kim, Sang-Wook Kang, Jandee Lee, Jong Ju Jeong, Woong Youn Chung, Daham Kim, Kee-Hyun Nam

**Affiliations:** 1Department of Surgery, CHA Ilsan Medical Center, CHA University School of Medicine, Goyang-si 10414, Korea or; 2Department of Surgery, Yonsei University College of Medicine, Seoul 03722, Korea; 3Department of Pathology, Yonsei University College of Medicine, Seoul 03722, Korea; 4Department of Internal Medicine, Institute of Endocrine Research, Yonsei University College of Medicine, Seoul 03722, Korea

**Keywords:** papillary thyroid cancer, TERT promoter, BRAF V600E, mutational analysis

## Abstract

**Simple Summary:**

TERT promoter mutation has recently emerged as a promising prognostic biomarker for aggressive papillary thyroid cancer (PTC), along with BRAF B600E mutation. The prevalence of the TERT promoter mutations has been reported as relatively uncommon in Asian countries. We report on a prospective study of the TERT promoter and BRAF V600E mutation in the largest number of subjects with PTC in Korea. We assume that our specific clinical settings and the favorable healthcare environment in Korea led to several distinct findings: the lowest prevalence of TERT promoter mutation ever reported, multifocal gene mutations in bilateral PTCs, and more early-stage papillary microcarcinomas included in this study. This study indicates that relevant evaluation and treatment strategies should be investigated continuously based on different circumstances.

**Abstract:**

Telomerase reverse transcriptase (TERT) promoter mutation has been investigated for its clinical and prognostic significance in aggressive papillary thyroid cancer (PTC). In this study, we aimed to assess the prevalence, clinicopathologic features, and treatment outcomes of TERT mutation-positive PTCs along with the common BRAF V600E mutation. We performed mutational analyses for BRAF and the TERT promoter in thyroid cancer patients who had undergone surgery at our institution since 2019. We reviewed and analyzed 7797 patients with PTC in this study. The prevalence of BRAF V600E and TERT promoter mutations was 84.0% and 1.1%, respectively. Multifocal gene mutations in bilateral PTCs were identified. TERT promoter mutations were associated with older age, larger tumor size, tumor multifocality, tumor variants, advanced stages, more adjuvant radioactive iodine treatment (RAI), higher stimulated serum thyroglobulin level before RAI, and more uptakes in the regions outside the surgical field on a post-RAI whole-body scan. The coexistence of BRAF V600E and TERT promoter mutations exacerbated all clinicopathologic characteristics. The frequency of TERT promoter mutations was the lowest in this study, compared to previous studies. TERT promoter mutations consistently correlated with aggressive PTCs, and the synergistic effect of both mutations was evident. Specific clinical settings in our institution and in Korea may have led to these distinctive results. Prospective multicenter studies with longer follow-up periods are required to establish valuable oncologic outcomes.

## 1. Introduction

Recently, the incidence of thyroid cancer has increased phenomenally worldwide, including in South Korea [1,2]. It is the most common type of cancer according to the 2019 annual report of cancer statistics in Korea (http://www.cancer.go.kr/, accessed on 17 May 2022), and papillary thyroid cancer (PTC) accounts for more than 95% of all thyroid cancer cases [2,3]. Regardless of a generally indolent disease course and good prognosis, 10–20% of PTC cases have aggressive features affecting frequent recurrence and higher mortality [4,5,6]. Thus, proper risk stratification and treatment for patients with aggressive PTC are crucial.

Several molecular markers have been investigated for accurately predicting disease prognosis [7,8,9,10]. The BRAF V600E mutation is the most prominent molecular marker of PTC, and is associated with aggressive clinicopathologic features and poor prognosis [11,12,13]. However, controversies in the clinical implications and the higher prevalence of the BRAF V600E mutation in Korean PTC patients have cast doubts on its true prognostic value [14,15,16].

The telomerase reverse transcriptase (TERT) gene has recently emerged as another promising prognostic biomarker for PTC. Mutations in its promoter region can re-activate telomerase, by triggering the unchecked replication of telomeric DNA, which can lead to cell immortalization and tumorigenesis [17,18]. Mutations in the TERT promoter region have been identified in several cancers, including thyroid cancer [19,20,21]. They frequently occur at two hotspots: −124 bp (C228T) and −146 bp (C250T) upstream of the ATG start codon, opening up new binding sites for E-twenty-six (ETS) transcription factors involved in TERT promoter hyperactivation.

Previous studies demonstrated a higher prevalence of TERT promoter mutations in aggressive thyroid cancer with poor clinicopathologic characteristics [22,23,24]. Their prevalence in PTC reportedly varies from 5.7–17.0%, prompting the investigation of their diagnostic and prognostic utility [25,26]. Furthermore, several studies have shown synergistic interactions with the BRAF V600E mutation to be associated with far more aggressive forms of PTC [27,28,29,30].

As a single tertiary center, mutational analyses for BRAF and TERT promoter in all thyroid cancer patients have been implemented in our institution since 2019. Firstly, this study aimed to estimate the empirical prevalence of TERT promoter mutations in consecutive cases of PTC. Secondly, it aimed to evaluate the association between TERT promoter and BRAF V600E mutations, their clinicopathologic features, and the treatment outcomes in patients with PTC.

## 2. Materials and Methods

### 2.1. Patients

A review of a prospectively designed database at our institute from January 2019 to December 2021 revealed 8962 consecutive patients with PTC underwent surgery at the Department of Surgery, Yonsei Cancer Center in Severance Hospital, Yonsei University College of Medicine. Thyroid surgery and adjuvant radioactive iodine (RAI) therapy were recommended based on the clinical findings. The routine process of preoperative work up, surgical procedure, and outcome assessment was established as previously described [31]. Patients were excluded if they refused mutational analysis or had the following: missing clinical, radiological, and/or pathological data; previous thyroid surgery; any prior cancer history. Clinical characteristics and demographic data were analyzed in 7797 patients who were finally included in the study (Figure 1). The demographic data are listed in Appendix A.

Benign, benign nodules; PTC, papillary thyroid cancer; MTC, medullary thyroid cancer; FTC, follicular thyroid cancer; ATC, anaplastic thyroid cancer; Wild, wild-type TERT; C228T, C228T mutation; C250T, C250T mutation

### 2.2. Histopathologic Diagnosis

The histopathologic diagnosis of PTC was confirmed by experienced pathologists, based on the World Health Organization’s current diagnostic criteria [32]. Papillary microcarcinoma (PMC) was diagnosed as PTC with tumor diameter ≤1 cm. Tumor–node–metastasis (TNM) staging was classified using the 8th edition of the American Joint Committee on Cancer (AJCC) staging system [33,34]. Based on this staging system, extrathyroidal extension of the tumor was defined as the tumor extending grossly to the strap muscles or other organs (≥T3b stage), which was confirmed by the operation and the pathology report.

### 2.3. Mutational Analyses

Mutational analyses of the BRAF gene and the TERT promoter were conducted on thyroid tumors from paraffin-embedded tissue blocks. In unilateral multifocal PTCs, analyses were performed on the tumor with the largest diameter, defined as the primary lesion. In patients with bilateral nodules of Bethesda categories V (suspicious for malignancy) or VI (malignancy), mutational analyses were conducted on those tumors from both lobes. The larger tumor was designated the “main” lesion, and the other tumor in the contralateral lobe was called the “contralateral” tumor.

Genomic DNA was extracted from 10-μm-thick formalin-fixed paraffin-embedded tissue blocks using the RecoverAll™ Total Nucleic Acid Isolation Kit (Life Technologies, Carlsbad, CA, USA). Tumor areas were manually dissected with a DNA-free scalpel under a microscope to obtain the significant portion of the tumor.

The presence of BRAF V600E and TERT promoter mutations was evaluated using pyrosequencing per the manufacturer’s instructions. BRAF V600E mutations were detected by polymerase chain reaction (PCR) amplification of exon 15 of the BRAF gene, using a previously reported forward primer (5′-biotin-TTCTTCATGAAGACCTC ACAGTAA-3′) and reverse primer (5′-CCAGACAACTGTTCAAACTGATG-3′), on a C1000 thermal cycler (BIO-RAD, California, USA). The pyrosequencing reaction was performed with a sequencing primer (5′-GGACCCACTCCCATCGAGATTT-3′) on a PyroMark Q24 instrument (Qiagen, Hilden, Germany). The TERT promoter was PCR-amplified to detect mutations in the common hotspots (C228T and C250T) using the forward primer (5′-CTTCACCTTCCAGCTCCG-3′) and reverse primer (5′-AAA GGAAGGGGAGGGGCTG-3′). The pyrosequencing reaction was executed with a sequencing primer (5′- CCCGCCCCGTCCCGA-3′). The produced pyrograms were analyzed with the PyroMark Q24 software version 2.0 (Qiagen, Valencia, CA, USA) to distinguish the mutant versus wild-type alleles by relative peak height [35].

### 2.4. Statistical Analyses

Only the main lesions were considered in the analyses, not the contralateral tumors. Categorical variables are presented as numbers and percentages, and continuous variables are presented as the mean ± standard deviation. Categorical variables were compared using the chi-squared (χ^2^) test, Fisher’s exact test, or linear-by-linear association when appropriate. Continuous variables were compared using Student’s *t*-test for two groups and analysis of variance for more than two groups. For multiple comparisons, a Bonferroni correction method was applied. *p*-values < 0.05 were considered statistically significant. All data were analyzed using IBM SPSS Statistics for Windows, version 25.0 (IBM Corp., Armonk, NY, USA).

The study was conducted in accordance with the 1964 Declaration of Helsinki and was approved by the Institutional Review Board of the Severance Hospital (No. 4-2022-0470, 7 June 2022). All patients provided written informed consent for all the perioperative procedures and the mutational analyses for PTCs.

## 3. Results

### 3.1. Prevalence of TERT Promoter and BRAF V600E Mutations in PTC

Among 7797 patients with PTC, TERT promoter mutations were found in 87 (1.1%), of which the C228T mutation was observed in 76 (1.0%) and the C250T in 11 (0.1%) (Figure 1). Meanwhile, the BRAF V600E mutation was observed in 6546 patients (84.0%). The coexistence of BRAF and TERT promoter mutations was identified in 70 patients (0.9%), while 1234 (15.8%) had no mutations (Table 1).

Mutational analyses were performed on both the main lesions and the contralateral tumors from 732 patients with bilateral PTCs. Among them, 549 (75.0%) had the BRAF V600E mutation in both lesions, 101 (13.1%) had it in the main tumor alone, and 40 (5.5%) in the contralateral tumor alone. TERT promoter mutations were detected in the main lesions of 13 patients (1.8%), and two (0.3%) of them had TERT mutation-positive contralateral tumors. The C228T mutation was found in 12 patients (1.6%), and the C250T in one (0.2%) in the main tumor alone (Figure 2).

### 3.2. Clinicopathologic Characteristics of PTCs with TERT Promoter Mutations

The relationship between TERT promoter mutations and the clinicopathologic characteristics of PTCs was evaluated (Table 2). These mutations were significantly associated with age ≥ 55 years (*p* < 0.001), male sex (*p* < 0.001), tumor size > 1 cm (*p* < 0.001), multifocal PTCs (*p* < 0.001), extrathyroidal extension (*p* < 0.001), PTC variants (*p* = 0.004), perinodal infiltration (*p* < 0.001), T3–4 (*p* < 0.001)/N1 (*p* = 0.009)/M1 (*p* < 0.001) stages, and stage III–IV (*p* < 0.001). More patients were involved in adjuvant RAI therapy after surgery (*p* < 0.001). No significant correlation was observed between TERT promoter mutations and BRAF V600E mutations (*p* = 0.372).

Since 5592 (71.7%) of the patients had PMCs, we performed further analyses in 2205 patients with tumor diameter > 1 cm. The analyses yielded similar results. TERT promoter mutations were more frequent in age ≥ 55 years (*p* < 0.001), male sex (*p* < 0.001), larger tumor size (*p* < 0.001), multifocal PTCs (*p* = 0.020), extrathyroidal extension (*p* < 0.001), perinodal infiltration (*p* < 0.001), T3–4 (*p* < 0.001)/M1 (*p* < 0.001) stages, stage III–IV (*p* < 0.001), and more adjuvant RAI therapy (*p* < 0.001). BRAF V600E mutation (*p* = 0.881), tumor variants (*p* = 0.120), and N1 stage (*p* = 0.442) showed no association with TERT promoter mutations (Table 2).

### 3.3. Clinicopathologic Characteristics of PTCs According to BRAF V600E and TERT Promoter Mutational Status

Clinicopathologic features of PTCs with either BRAF or TERT promoter mutations and with both mutations were individually compared with those without mutations (Table 3). In the analysis of all PTCs, BRAF V600E mutation alone was more frequently observed in male sex (*p* < 0.001), tumor size ≤ 1 cm (*p* < 0.001), classic PTCs (*p* < 0.001), no perinodal infiltrations (*p* = 0.005), and N1 (*p* = 0.009) and M0 (*p* < 0.001) stages. TERT promoter mutations alone were significantly associated with age ≥ 55 years (*p* = 0.001), tumor size > 1 cm (*p* < 0.001), multifocal PTCs (*p* < 0.001), extrathyroidal extension (*p* < 0.001), PTC variants (*p* = 0.001), T3–4 (*p* < 0.001)/M1 stages (*p* < 0.001), stage III/IV (*p* < 0.001), and more adjuvant RAI therapy (*p* < 0.001). In contrast, the coexistence of both mutations was strongly associated with all the aggressive clinicopathologic characteristics followed by more adjuvant RAI therapy.

Comparisons among the PTC patients with tumor diameter > 1 cm revealed mostly similar findings, but different associations were seen with some clinicopathologic features. BRAF V600E mutation alone significantly correlated with extrathyroidal extension (*p* < 0.001), but showed an insignificant relationship with perinodal infiltrations (*p* = 0.174). TERT promoter mutations alone lost their significant association with PTC variants (*p* = 0.089). The coexistence of both mutations maintained its association with aggressive characteristics, except for tumor size (*p* = 0.666), multifocality (*p* = 0.129), N1 (*p* = 0.136), and M1 stages (*p* = 0.124) (Table 3).

### 3.4. Clinical Significance of Coexisting BRAF V600E and TERT Promoter Mutations in PTC

We further investigated the impact of coexisting BRAF V600E and TERT promoter mutations by comparing them with the “no mutation” and “BRAF V600E mutation alone” groups (Table 4). PTCs with both mutations showed the worst clinicopathologic findings for all the features, except perinodal infiltration (*p* = 0.056) and M stages (*p* = 0.358). Similar findings were observed in PTC with tumor diameter >1 cm, except for the loss of significant association with tumor multifocality (*p* = 0.080).

Compared with the “BRAF V600E mutation alone” group, the genetic duet of BRAF V600E and TERT promoter mutations showed markedly aggressive features. Patients were much older (in their sixties), and more male patients were involved. The double mutation group was significantly associated with tumor size > 1 cm, multifocal PTCs, extrathyroidal extension, perinodal infiltration, T3–4/N1/M1 stages, and stage III–IV. The severe aggressiveness of these cancers necessitated more adjuvant RAI therapy. Analyses of PTC with tumor diameter > 1 cm showed corresponding results, except statistically insignificant associations with tumor multifocality and N1 stages (Table 4).

### 3.5. Assessment of Oncologic Outcomes after Adjuvant RAI Therapy

Out of the total number of patients, adjuvant RAI therapy was performed in 1486 (19.1%). Among the 7710 patients without TERT promoter mutations, 1432 (18.6%) were marked for adjuvant RAI treatment, while a majority of patients with TERT promoter mutations (54/87, 62.1%) were prescribed this treatment course. Analyses revealed that TERT promoter mutations were associated with higher stimulated thyroglobulin (Tg) levels before RAI (*p* = 0.020), and more uptakes in the cervical lymph nodes (LNs) or in the extracervical area on a post-therapeutic whole-body scan (WBS) (*p* = 0.001). The levels of serum thyroid-stimulating hormone (TSH) (*p* = 0.830) and Tg antibody (Ab) (*p* = 0.443) before RAI were similar. No differences were observed in the levels of serum TSH (*p* = 0.522), unstimulated Tg (*p* = 0.095), and Tg Ab (*p* = 0.463) between the TERT mutation-positive and wild-type TERT groups at 6 months after RAI. The mean serum TSH levels in the TERT mutation-positive group were low (0.32 uIU/mL), while those in the wild-type TERT group were higher (1.02 uIU/mL). A few TERT mutation-positive patients had extremely high serum unstimulated Tg levels (up to several thousands), but this was statistically insignificant (Table 5).

Analyses of 873 PTC patients with tumor diameter > 1 cm showed similar findings. Patients with TERT promoter mutations showed elevated stimulated Tg levels before RAI (*p* = 0.027) and more uptakes in the cervical LNs or in the extracervical area on post-therapeutic WBS (*p* = 0.025). The levels of serum TSH (*p* = 0.591) and Tg Ab (*p* = 0.526) before RAI and those 6 months after RAI (*p* = 0.552, *p* = 0.422, respectively) and unstimulated Tg at 6 months after RAI (*p* = 0.117) did not significantly differ according to the TERT promoter mutational status (Table 5).

## 4. Discussion

With the rising incidence of PTC and the development of molecular biomarkers in recent decades, numerous studies have reported the prevalence of BRAF V600E and TERT promoter mutations in PTC. The incidence of the BRAF V600E mutation is much higher than that of the uncommon TERT promoter mutations in Asian countries, with the former reported in 80.8–88.2% of all cases and the latter in only 2.8–5.7% [25,36,37,38].

We found BRAF V600E in 84.0% of our cases, similar to the frequency reported in other Asian countries. However, the TERT promoter mutations were detected in only 1.1% of PTC patients, the lowest prevalence reported in the literature. Most subjects in this study harbored indolent PTCs: 71.7% had PMCs, 80.2% were less than 55 years old, and 85.8% had stage I cancers. This phenomenon may be due to the accessibility of health check-up services, the referral pattern in Korea, and ethnicity [2]. Another possible explanation is that mutational analyses are performed for all thyroid cancer patients. Being one of the largest tertiary referral centers in Korea, our experience is from several thousands of patients per year, and the routine mutational analyses can lead to even lower disease prevalence. Based on these observations, we further analyzed only those PTCs with tumors larger than 1 cm. Incidence of TERT promoter mutations in these patients was 3.2%, which is more comparable to previous reports.

This is the first report on multifocal gene mutations in bilateral PTCs. Most bilateral cancers had bilateral BRAF V600E mutations, both in the main lesions as well as contralateral tumors, while a few harbored this mutation in the contralateral tumors only. However, bilateral TERT promoter mutations were extremely rare. Although limited to patients preoperatively diagnosed with bilateral PTCs, these findings substantially emphasize the stark difference in the frequencies of these two mutations. The impact of gene mutations in either or both sides of bilateral PTCs on the clinical behaviors and disease prognosis needs further investigation.

In accordance with the previous studies, our study showed that TERT promoter mutations are generally associated with aggressive clinicopathologic characteristics of PTC [22,24]. However, the BRAF V600E mutation did not show significant correlation with the TERT promoter mutations. The association between the two genes has been investigated, reporting diverse results. The lack of correlation between the two mutations may be, in part, due to their dramatically different frequencies in the population [25].

Analyses were based on the mutational status of both BRAF and TERT promoter since more patients had the double mutation. TERT promoter mutation alone was associated with aggressive clinicopathologic characteristics: old age, larger tumor size, multifocal PTCs, extrathyroidal extensions, PTC variants, perinodal infiltration, distant metastasis, and stage III–IV. These are mostly consistent with previous studies [26,39], except for nodal metastasis. The insignificance of nodal metastasis may follow from our strategy of treating all patients, either prophylactically or therapeutically, with routine central compartment neck dissection (CCND). In fact, node-negative patients were equally prevalent in the BRAF V600E, TERT promoter mutation, or no mutation groups. Therapeutic CCND could have produced more differences in nodal aggressiveness between the groups only in PTCs with clinically apparent nodal metastases.

BRAF V600E mutation alone was strongly associated with male sex, smaller tumor size, classic PTCs, less perinodal infiltrations, and N1, M0 stages of PTCs. Although this mutation is well known for its contribution to tumor aggressiveness, there are a few controversies on the link between this mutation and PTC aggressiveness [14,15]. Moreover, the higher prevalence of early PMCs in this study can weaken its association with aggressive characteristics, such as extrathyroidal extension of the tumor. Further evaluation in larger group, multicenter studies with long-term follow-up is necessary to highlight its clinical significance in PTC.

Compared with either BRAF or TERT promoter mutations alone, the coexistence of both mutations was significantly related to far more aggressive PTCs: older age, male sex, larger tumor size, tumor multifocality, extrathyroidal extension, perinodal infiltration, and more advanced T/N/M stages were predominant. Previous studies have also demonstrated the synergistic effect of BRAF V600E and TERT promoter mutations on exacerbating the clinicopathologic features of PTC [9,27,30,37]. This effect is initiated by the BRAF-induced activation of the mitogen-activated protein kinase (MAPK) pathway, which upregulates the ETS transcription factors. They bind to the increased ETS-binding sites on the mutant TERT promoter resulting in escalated TERT mRNA expression [40].

The clinical relevance of TERT promoter mutations in PTC aggressiveness was also reflected in several treatment outcomes. Patients with these mutations received more adjuvant RAI ablation. They showed higher serum-stimulated Tg levels before RAI, and more uptakes in the cervical LNs or in the extracervical area on the post-therapeutic WBS. These findings correlate with those of previous studies that have demonstrated elevated levels of stimulated Tg, poor treatment response, and RAI refractoriness in TERT mutation-positive PTCs [41]. Serum TSH and unstimulated Tg levels 6 months after RAI were not statistically different according to the TERT promoter mutational status. However, numerically low TSH (<1.0 uIU/mL) and higher unstimulated Tg levels after RAI were observed in TERT mutation-positive cases, implying more application of TSH-suppression therapy in these patients for poorer treatment response after RAI. Our study contained a short follow-up period, and there are limitations to studying long-term oncologic outcomes such as recurrence. These values should be investigated in future studies.

Compared with previous studies, this study has several distinct features that can be advantages as well as limitations. First, this study included the largest group of subjects ever reported in PTC research. Several thousands of patients undergo surgery every year, and routine mutational analyses were available in our institution. Second, multifocal mutations of these genes were verified for the first time, through mutational analyses in bilateral cancers. In this study, we evaluated only preoperatively diagnosed bilateral PTCs, and not all pathologically confirmed bilateral PTCs were included. Substantial research is needed to evaluate their clinical significance in larger cohorts. Third, despite initially being designed as a prospective cohort study, data were retrospectively reviewed and from a single institution. Fourth, this study was based on the Korean population, and the results cannot be extrapolated to patients of other ethnicities. Well-developed health check-up services and the referral pattern in Korea can lead to the inclusion of more early-stage PMCs in the study. Fifth, the follow-up period was short, and the outcome assessments were limited. Sixth, the cost-effectiveness of the mutational analyses had not been considered. Patients with PMC are predominant in this study, and the incidence of TERT promoter mutation in these patients is even rarer (0.29%). The scanty TERT promoter mutations in PMCs are concordant findings with the previous studies, and this can indicate the lower clinical significance of TERT promoter mutation in the clinical field [42]. This can raise questions about the necessity of routine mutational analysis, and relevant evaluation and treatment strategies should continuously be investigated.

## 5. Conclusions

In summary, we report our experience of TERT promoter mutations along with BRAF V600E in the largest cohort study of PTC patients in Korea. The BRAF V600E mutation was highly frequent, as reported in other Asian countries, but TERT promoter mutations had the lowest prevalence ever reported (1.1%). Multifocal gene mutations in bilateral PTCs were identified for the first time. The TERT promoter mutations were associated with aggressive clinicopathologic features of PTCs, while the clinical significance of BRAF V600E in affecting PTC aggressiveness was diminished. We assume that the specific clinical settings of our evaluation and treatment strategies and the favorable healthcare environment in Korea led to the identification of many early-stage PMCs in this study, resulting in several distinctive findings. The synergistic effect of both genes was obvious, playing a crucial role in exacerbating clinicopathologic features of PTC. The short follow-up period is a significant limitation to evaluating the relevant treatment outcomes. A prospective, multicenter study with long-term follow-up is essential to validate the therapeutic and prognostic implications of these genes.

## Figures and Tables

**Figure 1 cancers-14-04928-f001:**
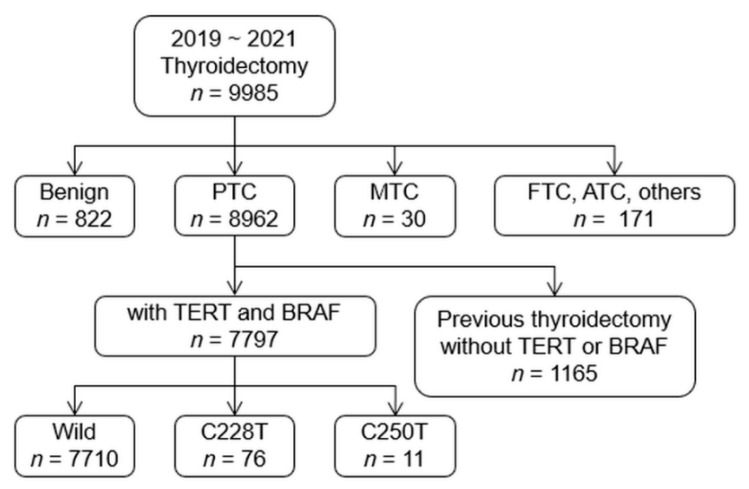
Initial process of patients’ selection for the study.

**Figure 2 cancers-14-04928-f002:**
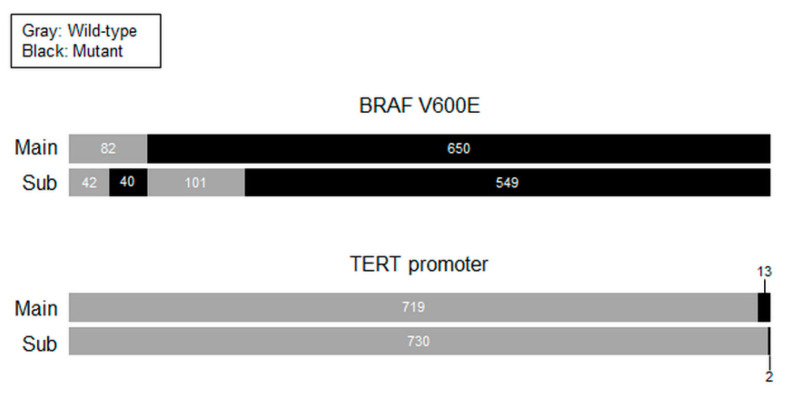
BRAF V600E and TERT promoter mutations in preoperatively diagnosed bilateral PTCs. Mutational status in main lesions and subtumors.

**Table 1 cancers-14-04928-t001:** Prevalence of TERT promoter and BRAF V600E mutations.

Variables	Wild-Type TERT	TERT Promoter Mutation C228T/C250T	Total
Wild-type BRAF	1234 (15.8)	16/1 (0.2)	1251 (16.0)
BRAF V600E	6476 (83.1)	60/10 (0.9)	6546 (84.0)
Total	7710 (89.9)	76/11 (1.1)	7797 (100.0)

Values are expressed as numbers (%).

**Table 2 cancers-14-04928-t002:** TERT promoter mutations and clinicopathologic features in patients with PTC.

	Total PTC			PTC > 1 cm		
Variables	Wild-Type TERT	pTERT Mutation	*p*-Value	Wild-Type TERT	pTERT Mutation	*p*-Value
Number of cases (%)	7710 (98.9)	87 (1.1)		2134 (96.8)	71 (3.2)	
Age (year)	43.04 ± 12.03	62.18 ± 12.17	<0.001	41.72 ± 12.96	63.04 ± 12.55	<0.001
≥55	1481 (19.2)	66 (75.9)	<0.001	383 (17.9)	54 (76.1)	<0.001
Sex, Male	1765 (22.9)	39 (44.8)	<0.001	591 (27.7)	34 (47.9)	<0.001
Tumor size (cm)	0.90 ± 0.67	1.95 ± 1.28	<0.001	1.69 ± 0.79	2.25 ± 1.24	<0.001
>1.0	2134 (27.7)	71 (81.6)	<0.001	-	-	-
Multifocality	2537 (32.9)	45 (51.7)	<0.001	877 (41.1)	39 (54.9)	0.020
Extrathyroidal extension	692 (9.0)	43 (49.4)	<0.001	515 (24.1)	40 (56.3)	<0.001
Histology						
Classic	7109 (92.2)	73 (83.9)	0.020	1875 (87.9)	58 (81.7)	0.184
Follicular variant	396 (5.1)	9 (10.3)		179 (8.4)	8 (11.3)	
Tall-cell variant	76 (1.0)	3 (3.4)		19 (0.9)	3 (4.2)	
Others	129 (1.7)	2 (2.3)		61 (2.9)	2 (2.8)	
Histology, Variants	601 (7.8)	14 (16.1)	0.004	259 (12.1)	13 (18.3)	0.120
BRAF V600E mutation	6476 (84.0)	70 (80.5)	0.372	1669 (78.2)	55 (77.5)	0.881
Perinodal infiltration	701 (9.1)	30 (34.5)	<0.001	412 (19.3)	26 (36.6)	<0.001
T stage						
T1	6760 (87.7)	34 (39.1)	<0.001	1361 (63.8)	21 (29.6)	<0.001
T2	234 (3.0)	8 (9.2)		234 (11.0)	8 (11.3)	
T3	501 (6.5)	27 (31.0)		390 (18.3)	27 (38.0)	
T4	215 (2.8)	18 (20.7)		149 (7.0)	15 (21.1)	
T stage, T3–T4	716 (9.3)	45 (51.7)	<0.001	539 (25.3)	42 (59.2)	<0.001
N stage						
N0	4275 (55.4)	36 (41.4)	<0.001	747 (35.0)	28 (39.4)	0.749
N1a	2687 (34.9)	29 (33.3)		912 (42.7)	22 (31.0)	
N1b	748 (9.7)	22 (25.3)		475 (22.3)	21 (29.6)	
N stage, N1	3435 (44.6)	51 (58.6)	0.009	1387 (65.0)	43 (60.6)	0.442
M stage, M1	15 (0.2)	8 (9.2)	<0.001	12 (0.6)	7 (9.9)	<0.001
TNM stage						
I	6535 (84.8)	26 (29.9)	<0.001	1661 (77.8)	19 (26.8)	<0.001
II	1124 (14.6)	40 (46.0)		439 (20.6)	34 (47.9)	
III	46 (0.6)	13 (14.9)		30 (1.4)	11 (15.5)	
IV	5 (0.1)	8 (9.2)		4 (0.2)	7 (9.9)	
TNM stage, III–IV	51 (0.7)	21 (24.1)	<0.001	34 (1.6)	18 (25.4)	<0.001
RAI therapy	1432 (18.6)	54 (62.1)	<0.001	824 (38.6)	49 (69.0)	<0.001

Values are expressed as the mean ± standard deviation or as numbers (%). PTC, papillary thyroid cancer; pTERT, TERT promoter; RAI, radioactive iodine.

**Table 3 cancers-14-04928-t003:** BRAF and TERT promoter mutations and clinicopathologic features in patients with PTC.

	Total PTC							PTC > 1 cm						
Variables	No Mutation	BRAF V600E Mutation Only	*p*- Value	pTERT Mutation Only	*p*- Value	BRAF+ pTERT Mutations	*p*- Value	No Mutation	BRAF V600E Mutation Only	*p*- Value	pTERT Mutation Only	*p*- Value	BRAF+ pTERT Mutations	*p*- Value
No. of cases	1234 (15.8)	6476 (83.1)		17 (0.2)		70 (0.9)		465 (21.1)	1669 (75.7)		16 (0.7)		55 (2.5)	
Age (year)	42.52 ± 12.56	43.14 ± 11.93	0.113	58.18 ± 14.22	<0.001	63.16 ± 11.52	<0.001	39.67 ± 12.71	42.29 ± 12.97	<0.001	58.75 ± 14.48	<0.001	64.29 ± 11.79	<0.001
≥55	240 (19.4)	1241 (19.2)	0.815	9 (52.9)	0.001	57 (81.4)	<0.001	64 (13.8)	319 (19.1)	0.008	9 (56.3)	<0.001	45 (81.8)	<0.001
Sex, Male	228 (18.5)	1537 (23.7)	<0.001	4 (23.5)	0.573	35 (50.0)	<0.001	108 (23.2)	483 (28.9)	0.015	4 (25.0)	0.772	30 (54.5)	<0.001
Tumor size (cm)	1.09 ± 0.95	0.86 ± 0.59	<0.001	2.92 ± 1.98	0.002	1.72 ± 0.93	<0.001	1.95 ± 1.05	1.62 ± 0.68	<0.001	3.06 ± 1.95	0.039	2.01 ± 0.83	0.666
>1.0	465 (37.7)	1669 (25.8)	<0.001	16 (94.1)	<0.001	55 (78.6)	<0.001	-	-	-	-	-		
Multiplicity	384 (31.1)	2153 (33.2)	0.145	13 (76.5)	<0.001	32 (45.7)	0.011	179 (38.5)	698 (41.8)	0.197	12 (75.0)	0.007	27 (49.1)	0.129
Extrathyroidal extension	95 (7.7)	597 (9.2)	0.087	6 (35.3)	<0.001	37 (52.9)	<0.001	80 (17.2)	435 (26.1)	<0.001	6 (37.5)	0.037	34 (61.8)	<0.001
Histology														
Classic	945 (76.6)	6164 (95.2)	<0.001	7 (41.2)	0.043	66 (94.3)	0.018	300 (64.5)	1575 (94.4)	<0.001	7 (43.8)	0.485	51 (92.7)	0.003
Follicular variant	223 (18.1)	173 (2.7)		9 (52.9)		0 (0.0)		124 (26.7)	55 (3.3)		8 (50.0)		0 (0.0)	
Tall-cell variant	1 (0.1)	75 (1.2)		0 (0.0)		3 (4.3)		0 (0.0)	19 (1.1)		0 (0.0)		3 (5.5)	
Others	65 (5.3)	64 (1.0)	<0.001	1 (5.9)	0.001	1 (1.4)	<0.001	41 (8.8)	20 (1.2)		1 (6.3)		1 (1.8)	
Histology, Variants	289 (23.4)	312 (4.8)		10 (58.8)		4 (5.7)		165 (35.5)	94 (5.6)	<0.001	9 (56.3)	0.089	4 (7.3)	<0.001
Perinodal infiltration	138 (11.2)	563 (8.7)	0.005	2 (11.8)	1.000	28 (40.0)	<0.001	100 (21.5)	312 (18.7)	0.174	2 (12.5)	0.541	24 (43.6)	<0.001
T stage														
T1	1028 (83.3)	5732 (88.5)	<0.001	6 (35.3)	<0.001	28 (40.0)	<0.001	274 (58.9)	1087 (65.1)	0.860	5 (31.3)	0.008	16 (29.1)	<0.001
T2	93 (7.5)	141 (2.2)		3 (17.6)		5 (7.1)		93 (20.0)	141 (8.4)		3 (18.8)		5 (9.1)	
T3	82 (6.6)	419 (6.5)		6 (35.3)		21 (30.0)		72 (15.5)	318 (19.1)		6 (37.5)		21 (38.2)	
T4	31 (2.5)	184 (2.8)		2 (11.8)		16 (22.9)		26 (5.6)	123 (7.4)		2 (12.5)		13 (23.6)	
T stage, T3–T4	113 (9.2)	603 (9.3)	0.864	8 (47.1)	<0.001	37 (52.9)	<0.001	98 (21.1)	441 (26.4)	0.019	8 (50.0)	0.006	34 (61.8)	<0.001
N stage														
N0	726 (58.8)	3549 (54.8)	<0.001	10 (58.8)	0.416	26 (37.1)	0.001	201 (43.2)	546 (32.7)	0.687	10 (62.5)	0.421	18 (32.7)	0.385
N1a	326 (26.4)	2361 (36.5)		2 (11.8)		27 (38.6)		128 (27.5)	784 (47.0)		1 (6.3)		21 (38.2)	
N1b	182 (14.7)	566 (8.7)		5 (29.4)		17 (24.3)		136 (29.2)	339 (20.3)		5 (31.3)		16 (29.1)	
N stage, N1	508 (41.2)	2927 (45.2)	0.009	7 (41.2)	0.999	44 (62.9)	<0.001	264 (56.8)	1123 (67.3)	<0.001	6 (37.5)	0.127	37 (67.3)	0.136
M stage														
M0	1225 (99.3)	6470 (99.9)	<0.001	13 (76.5)	<0.001	66 (94.3)	0.004	456 (98.1)	1666 (99.8)	<0.001	12 (75.0)	<0.001	52 (94.5)	0.124
M1	9 (0.7)	6 (0.1)		4 (23.5)		4 (5.7)		9 (1.9)	3 (0.2)		4 (25.0)		3 (5.5)	
TNM stage														
I	1059 (85.8)	5476 (84.6)	0.435	6 (35.3)	<0.001	20 (28.6)	<0.001	383 (82.4)	1278 (76.6)	0.005	6 (37.5)	<0.001	13 (23.6)	<0.001
II	170 (13.8)	954 (14.7)		5 (29.4)		35 (50.0)		79 (17.0)	360 (21.6)		4 (25.0)		30 (54.5)	
III	4 (0.3)	42 (0.6)		2 (11.8)		11 (15.7)		2 (0.4)	28 (1.7)		2 (12.5)		9 (16.4)	
IV	1 (0.1)	4 (0.1)		4 (23.5)		4 (5.7)		1 (0.2)	3 (0.2)		4 (25.0)		3 (5.5)	
TNM stage, III–IV	5 (0.4)	46 (0.7)	0.226	6 (35.3)	<0.001	15 (21.4)	<0.001	3 (0.6)	31 (1.9)	0.091	6 (37.5)	<0.001	12 (21.8)	<0.001
RAI therapy	250 (20.3)	1182 (18.3)	0.097	12 (70.6)	<0.001	42 (60.0)	<0.001	167 (35.9)	657 (39.4)	0.176	12 (75.0)	0.003	37 (67.3)	<0.001

Values are expressed as the mean ± standard deviation or as numbers (%). PTC, papillary thyroid cancer; pTERT, TERT promoter; RAI, radioactive iodine.

**Table 4 cancers-14-04928-t004:** The impact of both BRAF and TERT promoter mutations on clinicopathologic features of PTC in comparison with no mutation and BRAF mutation alone.

	Total PTC				PTC >1 cm			
Variables	No Mutation	BRAF Mutation Only	BRAF + pTERT Mutations	*p*-Value	No Mutation	BRAF Mutation Only	BRAF + pTERT Mutations	*p*-Value
No. of cases	1234 (15.9)	6476 (83.2)	70 (0.9)		465 (21.2)	1669 (76.2)	55 (2.5)	
Age (year)	42.52 ± 12.56	43.14 ± 11.93	63.16 ± 11.52 *^, #^	<0.001	39.67 ± 12.71	42.29 ± 12.97 *	64.29 ± 11.79 *^, #^	<0.001
≥55	240 (19.4)	1241 (19.2)	57 (81.4) *^, #^	<0.001	64 (13.8)	319 (19.1) *	45 (81.8) *^, #^	<0.001
Sex, Male	228 (18.5)	1537 (18.5) *	35 (50.0) *^, #^	<0.001	108 (23.2)	483 (28.9) *	30 (54.5) *^, #^	<0.001
Tumor size (cm)	1.09 ± 0.95	0.86 ± 0.59 *	1.72 ± 0.93 *^, #^	<0.001	1.95 ± 1.05	1.62 ± 0.68 *	2.01 ± 0.83 ^#^	<0.001
>1.0	465 (37.7)	1669 (25.8) *	55 (78.6) *^, #^	0.004	-	-	-	-
Multifocality	384 (31.1)	2153 (33.2)	32 (45.7) *^, #^	0.017	179 (38.5)	698 (41.8)	27 (49.1)	0.080
Extrathyroidal extension	80 (17.2)	435 (26.1)	37 (52.9) *^, #^	<0.001	80 (17.2)	435 (26.1)	34 (61.8) *^, #^	<0.001
Histology								
Classic	945 (76.6)	6164 (95.2) *	66 (94.3) *	<0.001	300 (64.5)	1575 (94.4) *	51 (92.7) *	<0.001
Follicular variant	223 (18.1)	173 (2.7)	0 (0.0)		124 (26.7)	55 (3.3)	0 (0.0)	
Tall-cell variant	1 (0.1)	75 (1.2)	3 (4.3)		0 (0.0)	19 (1.1)	3 (5.5)	
Others	65 (5.3)	64 (1.0)	1 (1.4)		41 (8.8)	20 (1.2)	1 (1.8)	
Histology, Variants	289 (23.4)	312 (4.8) *	4 (5.7) *	<0.001	165 (35.5)	94 (5.6) *	4 (7.3) *	<0.001
Perinodal infiltration	138 (11.2)	563 (8.7) *	28 (40.0) *^, #^	0.056	100 (21.5)	312 (18.7)	24 (43.6) *^, #^	0.062
T stage								
T1	1028 (83.3)	5732 (88.5) *	28 (40.0) *^, #^	<0.001	274 (58.9)	1087 (65.1) *	16 (29.1) *^, #^	<0.001
T2	93 (7.5)	141 (2.2)	5 (7.1)		93 (20.0)	141 (8.4)	5 (9.1)	
T3	82 (6.6)	419 (6.5)	21 (30.0)		72 (15.5)	318 (19.1)	21 (38.2)	
T4	31 (2.5)	184 (2.8)	16 (22.9)		26 (5.6)	123 (7.4)	13 (23.6)	
T stage, T3-T4	113 (9.2)	603 (9.3)	37 (52.9) *^, #^	<0.001	98 (21.1)	441 (26.4) *	34 (61.8) *^, #^	<0.001
N stage								
N0	726 (58.8)	3549 (54.8) *	26 (37.1) *^, #^	0.242	201 (43.2)	546 (32.7) *	18 (32.7)	0.376
N1a	326 (26.4)	2361 (36.5)	27 (38.6)		128 (27.5)	784 (47.0)	21 (38.2)	
N1b	182 (14.7)	566 (8.7)	17 (24.3)		136 (29.2)	339 (20.3)	16 (29.1)	
N stage, N1	508 (41.2)	2927 (45.2) *	44 (62.9) *^, #^	<0.001	264 (56.8)	1123 (67.3) *	37 (67.3) *	0.001
M stage								
M0	1225 (99.3)	6470 (99.9) *	66 (94.3) *^, #^	0.358	456 (98.1)	1666 (99.8) *	52 (94.5) ^#^	0.784
M1	9 (0.7)	6 (0.1)	4 (5.7)		9 (1.9)	3 (0.2)	3 (5.5)	
TNM stage								
I	1059 (85.8)	5476 (84.6)	20 (28.6) *^, #^	<0.001	383 (82.4)	1278 (76.6) *	13 (23.6) *^, #^	<0.001
II	170 (13.8)	954 (14.7)	35 (50.0)		79 (17.0)	360 (21.6)	30 (54.5)	
III	4 (0.3)	42 (0.6)	11 (15.7)		2 (0.4)	28 (1.7)	9 (16.4)	
IV	1 (0.1)	4 (0.1)	4 (5.7)		1 (0.2)	3 (0.2)	3 (5.5)	
TNM stage, III–IV	5 (0.4)	46 (0.7)	15 (21.4) *^, #^	<0.001	3 (0.6)	31 (1.9)	12 (21.8) *^, #^	<0.001
RAI therapy	250 (20.3)	1182 (18.3)	42 (60.0) *^, #^	0.004	167 (35.9)	657 (39.4)	37 (67.3) *^, #^	<0.001

Values are expressed as the mean ± standard deviation or as numbers (%). PTC, papillary thyroid cancer; pTERT, TERT promoter; RAI, radioactive iodine. * Significantly different from the no mutation group; ^#^ significantly different from the BRAF mutation only group.

**Table 5 cancers-14-04928-t005:** TERT promoter mutation in patients with PTC and adjuvant radioactive iodine therapy.

	Total PTC			PTC > 1 cm		
Variables	Wild-Type TERT	pTERT Mutation	*p*-Value	Wild-Type TERT	pTERT Mutation	*p*-Value
Number of adjuvant RAI cases	1432 (96.4)	54 (3.6)		824 (94.4)	49 (5.6)	
TSH level before RAI (uIU/mL)	75.50 ± 42.51	77.47 ± 66.65	0.830	74.07 ± 43.85	79.49 ± 69.34	0.591
Stimulated Tg before RAI (mg/mL)	15.75 ± 87.19	422.74 ± 1247.07	0.020	23.06 ± 110.95	446.77 ± 1302.63	0.027
Tg Ab before RAI (IU/mL)	60.78 ± 256.97	88.21 ± 287.04	0.443	63.12 ± 279.15	89.28 ± 297.97	0.526
Results of post-therapeutic WBS						
Thyroid bed only uptake	1318 (92.0)	43 (79.6)	0.001	740 (89.8)	39 (79.6)	0.025
Cervical LN or extracervical uptake	114 (8.0)	11 (20.4)		84 (10.2)	10 (20.4)	
Number of cases at 6 months after RAI	1076 (72.4)	40 (2.7)		605 (69.3)	35 (4.0)	
TSH (uIU/mL)	1.02 ± 6.96	0.32 ± 0.79	0.522	1.12 ± 7.60	0.35 ± 0.84	0.552
Unstimulated Tg (mg/mL)	1.10 ± 11.912	80.14 ± 291.90	0.095	1.70 ± 15.77	86.12 ± 310.80	0.117
Tg Ab (IU/mL)	42.11 ± 200.81	18.79 ± 42.55	0.463	52.97 ± 261.33	17.42 ± 42.92	0.422

Values are expressed as the mean ± standard deviation or as numbers (%). PTC, papillary thyroid cancer; pTERT, TERT promoter; RAI, radioactive iodine; Tg, thyroglobulin; Ab, antibody; WBS, whole-body scan; LN, lymph node.

## Data Availability

Some or all data generated or analyzed during this study are included in this published article or in the data repositories listed in the References.

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
