# Peer review of "TERT Promoter and BRAF V600E Mutations in Papillary Thyroid Cancer: A Single-Institution Experience in Korea"

_cancers, 2022, doi:10.3390/cancers14194928_

Round 1

Reviewer 1 Report

This study evaluated BRAF V600E and TERT promotor mutations in a large series of 7797 patients with papillary thyroid carcinoma. The study is well performed and I would like to congratulate the authors for putting this result together. Some minor suggestions are as follows. 

1. The term "subtumor" is pretty confusing. Please consider using a more descriptive term such as "main tumor v.s. multifocal smaller tumors".

2. Table 3 is a bit too busy to read. Please consider editing with smaller font size.

3. It is worth mentioning that 72% of the cases were microcarcinoma (<1 cm) in this study cohort and to discuss the differences between size and molecular profile in more detail, such as compared with prior literature, in the discussion section. 

Reviewer 2 Report

The main aim of this article entitled „TERT promoter and BRAF V600E mutations in papillary thyroid cancer: a single - institution experience in Korea“ was to evaluate the impact of the coexistence of BRAF V600E and TERT promoter mutations on clinicopathological characteristics of papillary thyroid carcinoma (PTC) and on its treatment outcome. The role of TERT promoter and BRAF V600E mutations in the pathogenesis and prognosis of PTC has been particularly clarified earlier, however little is known about the synergic effect of these mutations. This study was partly prospectively, partly retrospectively carried in a series of 7.797 cases of PTC in a single tertiary center. All laboratory and statistical methods used in this study are relevant and well described.

The authors confirmed the final effect of the coexistence of BRAF and TERT mutations related to the higher aggressiveness of PTC. This study seems me to be exceptional in analysis of the largest cohort of bilateral PTC. In the „Discussion“ the authors listed some limitations and shortcomings of this study and they suggest further prospective, multi-center study with long-term follow-up to clarify still unanswered questions.

The acceptance of this manuscript after minor revision is recommended. The publication of this paper would be desirable in the opinion of the reviewer.

The following points should be clarified in the article:

Minor points:

Simple Summary:

Page 1/Line 22/23: microcarcinomas……..the first  -c-  is missing

Results:

1.Page 4/Line 150

Should be there written… 60 patients (0.9%)…..in agreement with Table 1 (given 70 patients is mistake)

2.Page 6-7/ Table 3

Table 3 is not entirely clear and transparent. The Table may be issued in landscape orientation.

Final evaluation:    In respect of all above mentioned shortcomings I recommend to accept the manuscript after minor revision.
